# The Effect of Ground Type on the Jump Performance of Adults of the Locust *Locusta migratoria manilensis*: A Preliminary Study

**DOI:** 10.3390/insects11040259

**Published:** 2020-04-23

**Authors:** Chao Wan, Rentian Cao, Zhixiu Hao

**Affiliations:** 1State Key Laboratory of Tribology, Tsinghua University, Beijing 100084, China; 2Department of Mechanics, School of Aerospace Engineering, Beijing Institute of Technology, Beijing 100081, China

**Keywords:** ground type, jump, locust, reaction force, kinematics, elevation angle

## Abstract

The jump performance of locusts depends on several physiological and environmental factors. Few studies have examined the effects of different ground types on the jump performance of locusts. Here, mature adult locusts (*Locusta migratoria manilensis*) were examined using a custom-developed measuring system to test their jump performance (including postural features, kinematics, and reaction forces) on three types of ground (sand, soil, and wood). Significant differences were primarily observed in the elevation angle at take-off, the tibial angle at take-off, and the component of the mass-specific reaction force along the aft direction of the insect body between wood and the other two ground types (sand and soil). Slippage of the tarsus and insertion of the tibia were often observed when the locusts jumped on sand and soil, respectively. Nevertheless, comparisons of the different parameters of jump initiation (i.e., take-off speed and mass-specific kinetic energy) did not reveal any differences among the three types of ground, indicating that locusts were able to achieve robust jump performance on various substrates. This study provides insights into the biomechanical basis of the locust jump on different types of ground and enhances our understanding of the mechanism underlying the locust jump.

## 1. Introduction

Locusts are some of the most famous insects for their jumping ability, as they can achieve velocities as high as 2.6 m/s, accelerations as high as 75 m/s^2^, and can cover dozens of times their body length in a single jump [1]. Their jumps serve several critical functions: to escape from predators, to achieve an initial velocity for flight, and provide a more rapid alternative to travel than crawling. The locust jump has been extensively studied, especially its postural control, the mechanics of the hind leg, patterns of muscle and motoneuron activity, and mechanisms of energy storage and release [2,3,4,5,6,7,8,9,10,11]. The action of jumping in locusts is fueled by their hind legs in the following steps: initial flexion of the tibiae, co-contraction of the flexor and extensor muscles, and rapid tibial extension after trigger activity [2]. A large amount of strain energy is stored by the deformed exoskeleton during muscle co-contraction (especially by the semi-lunar process (SLP) cuticle on the distal end of the metathoracic femur) and is released during tibial extension to overcome the weaknesses of insect muscles [3,4].

Various physiological factors have been shown to have different effects on the jumping performance of locusts. Compared with immature juvenile hoppers, adult locusts have three times the range of escape jumps and at least twice the specific energy output [12,13]. Older juvenile insects hop less frequently than younger ones within the same instar because of increased anaerobic metabolism and locomotory fatigue [14]. Adaptive change in muscle contraction has also been observed in newly molted locusts to avoid cuticle damage during jumping [5]. In contrast, gravid females (20% heavier) have the same jump distance and significantly lower endurance compared with non-gravid females, as non-gravid females show a 20% increase in the duration of muscle contraction relative to gravid females [15,16]. Furthermore, Katz and Gosline [17] found that the take-off speed of the locust jump is relatively scale-independent (0.9–1.2 m/s for juveniles and 2.5 m/s for adults), showing that juvenile insects often jump to maximize the distance traveled while the purpose of jumping in adults more often serves to achieve a velocity necessary for initiating flight.

In addition, the effects of a few natural environmental factors have also been investigated. Hawlena et al. [18] reported that the chronic risk of predation can increase both the take-off speed and the jump distance of grasshoppers and that this pattern cannot be explained by morphological variation. Air resistance reduces the kinetic energy of locust jumps by less than 10% at lower initial speeds [19], and environmental temperatures ranging from 15 °C to 35 °C only weakly affect jump energy [20]. The properties of the ground are another set of environmental factors that can potentially affect the locust jump, especially during the take-off stage. For example, surface roughness with an Ra value of 1–2 μm can reduce the ability of locust legs to attach to the substrate, resulting in considerable slippage of the hind legs on the ground and thus take-off failure [21,22]. Similar effects of surface roughness have also been documented in females of the Mediterranean field cricket (*Gryllus bimaculatus*) crawling on smooth surfaces ($R_{q}$ = 7.3 μm), which resulted in significantly lower phonotactic responses compared with rougher surfaces ($R_{q}$ = 16 or 180 μm) [23]. Several natural types of ground surfaces have other key physical properties as well as surface roughness, such as normal stiffness, hardness, and tangential friction/shear stress strength. However, whether natural ground types can affect the jump performance of locusts has not been explored.

Here, we compared the kinematics, insect posture, and reaction force of the jump performance of adult locusts on three natural ground types (sand, soil, and wood). We hypothesized that the jump performance of locusts would differ among the three types of ground. Our results provide insight into the biomechanical basis of the locust jump on different types of ground and enhance our understanding of the mechanisms underlying the locust jump.

## 2. Materials and Methods

Mature Oriental migratory locusts (*Locusta migratoria manilensis*) were purchased from the Jiyuan locust-breeding facility in Anhui Province of China. Before tests, all locusts were fed with wheat leaves under natural light and room temperature (23–30 °C) for at least 2 weeks to ensure fully sclerotized cuticles after their final molt. 

A custom-developed test system was built to simultaneously measure both the kinematics and reaction force of the locust jumps. Figure 1 shows the components of the test system, including a coordinate background plate (indicated by the letter G), two high-speed cameras (CR600 × 2, Optronis GmbH, Kehl, Germany; Hero4, GoPro Inc., San Mateo, CA, USA; indicated by letters D and E), a video camera (IXUS 210, Canon Inc., Tokyo, Japan; indicated by the letter F), a desktop for data acquisition (indicated by the letter H), and two custom-designed platforms (indicated by the letter B) supported by a stand column and a three-dimensional high-precision force sensor (S3-001NTO-003, Bio-inspired Technology Corp., Nanjing, China; indicated by the letter A). The two platforms were set in the same horizontal plane with 1 mm of space. The two high-speed cameras were placed on the side of the platforms. The GoPro camera (120 fps) was used to obtain the jump trajectory of the animals, and the Optronis camera (1000 fps) was used to record the rapid movement of their hind legs before take-off. The Canon camera was placed above the system to take images of insect posture and the jump azimuth. The force sensor was used to measure the three-dimensional reaction force from one hind leg of the locust during jumping (full-scaled range of 1 N, resolution of 1 mN, and sample frequency of 10 kHz).

Three ground types (sand, soil, and wood) were studied. First, balsa wood was fixed on the surface of the platforms to simulate wooden ground. The balsa wood was cut from some timbers and polished using moderate sandpaper (#: P400). All obvious visual burrs were removed from the wood surface. The locust was prepared by attaching its wings together using adhesive tape and placing it on the platforms such that its left and right hind legs each stood on one platform. Either a sudden sound or the touch of a brush was used to trigger the locust’s escaping jump. Data were excluded for individuals that had two hind legs on the platform of the force sensor. Next, the balsa wood was replaced by two boxes (width × length × depth = 15 mm × 40 mm × 10 mm) that were filled with local sand (grain size of approximately 0.4 mm, density = 1.27 g/cm³) to create a sandy substrate or commercial potting soil (soft and rich in organic matter, density = 0.317 g/cm³) to create a soil substrate. Both of the fillings were lightly compacted, and the surface was carefully flattened before tests. Jumps of the locusts on the sand and the soil were tested following the aforementioned procedures. In total, nine adult locusts (4 females and 5 males) were included (body mass = 1.8 ± 0.51 g (mean ± SD)) and used for all the three ground types in this study. For avoiding the animal fatigue, only one ground type was tested in one day. For each ground type, each animal was tested less than seven trials and its first successful jump was selected to represent the jump performance of this individual. In brief, nine jumps were included for each ground type (one jump from each animal for each ground type). The sand, soil, and wood ground types corresponded to a granular substrate with weak cohesion, a granular substrate with strong cohesion, and a solid substrate, respectively.

The yaw angle of a jump (φ) was obtained from the top camera image as the angle between the body axis of the locust and the direction *X* of the force sensor (Figure 2). The real distance of each locust jump ($S_{d}$) was then obtained by correcting the camera recordings using the following equation: (1)$$S_{d}=\frac{{\bar{S}}_{d}}{\cos(\varphi )}$$
where ${\bar{S}}_{d}$ is the distance recordings from the lateral images. The physiological components of the reaction force for the locusts were calculated from the force measurements of the sensor using the following equation:(2)$${\bar{F}}_{a}=F_{x}\cos(\varphi )-F_{y}\sin(\varphi ),{\bar{F}}_{l}=F_{x}\sin(\varphi )+F_{y}\cos(\varphi ),{\bar{F}}_{n}=F_{z}$$
where $F_{x}$, $F_{y}$, and $F_{z}$ are the measurements from the force sensor along its axes X, Y, and Z, respectively. ${\bar{F}}_{a}$, ${\bar{F}}_{l}$, and ${\bar{F}}_{n}$ are the physiological components of the reaction force along the aft, lateral, and normal directions of the locust, respectively (Figure 2b). To eliminate the effect of body mass, the mass-specific reaction force was calculated by normalizing the physiological components with body mass as follows:(3)$$F_{a}=\frac{{\bar{F}}_{a}}{M},F_{l}=\frac{\left|{\bar{F}}_{l}\right|}{M},F_{n}=\frac{{\bar{F}}_{n}}{M},F_{t}=\sqrt{F_{a}^{2}+F_{l}^{2}+F_{n}^{2}}$$
where M is body mass. $F_{a}$, $F_{l}$, and $F_{n}$ are the aft, lateral, and normal components of the mass-specific reaction force, respectively. $F_{t}$ is the total magnitude of the mass-specific reaction force. The elevation angle of the locust at take-off ($\beta_{t}$) was determined based on the real trajectory of the jump from the images of the lateral GoPro camera after take-off (Figure 2).

Three postural features of the locust during jumps were defined based on the camera images as follows (Figure 3). First, the opening angle between the hind femur and the central line of the body (θ) was measured from the image of the top camera just before tibial extension. Following a previous study [6], a line was drawn through the proximal femoral and the distal tibial ends of the locust just before tibial extension, and its tilt angle relative to the ground surface was defined as $\beta_{e}$. Here, the angle $\beta_{e}$ was determined by both the projected tilt angle in the Optronis camera image (${\bar{\beta }}_{e}$) and the yaw angle φ as $\arctan(\cos(\varphi )\tan({\bar{\beta }}_{e}))$. Similarly, the angle between the tibial axis and the ground surface at take-off (γ) was calculated as $\arctan(\cos(\varphi )\tan(\bar{\gamma }))$, where $\bar{\gamma }$ is the projected tibia–ground angle in the Optronis camera image. Given the small waste in energy because of air resistance [19], the take-off speed ($V_{t}$) was calculated based on the real jump distance $S_{d}$ and the angle $\beta_{t}$ as
(4)$$V_{t}=\sqrt{\frac{gS_{d}\tan(\beta_{t})}{2}+\frac{gS_{d}}{2\tan(\beta_{t})}}$$
where g is the acceleration of gravity (i.e., 9.8 m s^−2^). The mass-specific kinetic energy ($E_{m}$) for each jump was determined as $0.5V_{t}^{2}$.

Statistical analyses were performed to clarify the effect of ground type on the locust jump. First, the normal distribution and homoscedasticity of the results were checked using Shapiro–Wilk tests and F-tests/Bartlett’s tests, respectively. If the data met these requirements, *t*-tests were performed for all of the data between female and male animals and between the angles $\beta_{t}$ and $\beta_{e}$ for each ground type. Repeated one-way analyses of variance (ANOVA) with a Bonferroni post-hoc correction were also used to compare results between the three different ground types. Mann–Whitney tests were used instead of *t*-tests if the data were not normally distributed, and *t*-tests with Welch’s correction were used if the data had unequal variances. For repeated one-way ANOVA, Friedman tests with Dunn’s multiple comparisons were used if the data were either not normally distributed or variances were not homogeneous. Results were reported as mean ± SD. Significance was defined as *p* < 0.05. 

## 3. Results

Both the postural features (θ, γ, $\beta_{t}$, and $\beta_{e}$) and jump performance ($V_{t}$ and $E_{m}$), including the reaction force (the maximal values of $F_{a}$, $F_{l}$, $F_{n}$, and $F_{t}$), of adult female locusts were not significantly (*p* > 0.05) different from those of adult male locusts regardless of ground type. As a result, data for both female and male locusts were pooled and analyzed together in subsequent statistical analyses. All of the experimental data are shown in Appendix A.

The four postural angles of the locust jumps on the three ground types are shown in Figure 4. The angles $\beta_{e}$ and $\beta_{t}$ were 57 ± 7.3 degrees and 36 ± 9.5 degrees, 52 ± 9.2 degrees and 43 ± 6.2 degrees, and 64 ± 12 degrees and 63 ± 12 degrees for the sand, soil, and wood substrates, respectively. Significant differences were detected in the angle $\beta_{e}$ between soil and wood (*p* < 0.05), in the angle $\beta_{t}$ between sand and wood (*p* < 0.001), and in the angle $\beta_{t}$ between soil and wood (*p* < 0.01). Comparisons between these two angles indicated that differences were only significant between sand and soil (*p* < 0.001). Moreover, the angle γ for the wood (68 ± 10 degrees) was significantly (*p* < 0.001) higher than that for the sand (44 ± 8.6 degrees) as well as that for the soil (51 ± 6.5 degrees). By contrast, the angle θ was not significantly affected by changes in ground type.

Figure 5 shows comparisons of the kinematics and reaction force among the three ground types. Neither the take-off speed ($V_{t}$) nor the mass-specific kinetic energy ($E_{m}$) was significantly different among the three ground types (2.3 ± 0.33 m/s and 2.7 ± 0.83 mJ/g for sand, 2.3 ± 0.13 m/s and 2.7 ± 0.31 mJ/g for soil, and 2.6 ± 0.17 m/s and 3.3 ± 0.42 mJ/g for wood). There were no significant differences in the mass-specific reaction force among the three ground types except for $F_{a}$. The $F_{a}$ for wood (20 ± 6.3 mN/g) was significantly (*p* < 0.05) lower than that for sand (38 ± 5.6 mN/g) and soil (31 ± 10 mN/g). In addition, the hind legs of the locusts during the jumps firmly adhered to the wood substrate, often inserted into the sandy substrate (six of the nine jumps), or noticeably slid on the soil substrate (seven of the nine jumps). To illustrate the different interactions between the hind legs of locusts and the three ground types, some typical high-speed camera images and videos of the locust hind legs during jumping are provided in Figure 6 and the Appendix A. 

## 4. Discussion

We suggest that locusts have a robust jump sequence on various substrates. Both the take-off speed and the mass-specific kinetic energy of the locust jump were not significantly affected by the change in ground type even though tibial slip or insertion occurred on sand and soil substrates during the initial stage of jumping (Figure 6b,c). Given that the main purpose of jumping for adult locusts is to achieve a sufficient initial speed for initiating flight [17], the fact that the take-off speed did not change among the different ground types revealed that the jump behaviors of the locust were robust on various substrates. The kinetic energy of the locust jump was primarily released from the elastic strain energy stored by the SLP cuticle during the extension of the hind leg [3,4]. The similar mass-specific kinetic energy for the jumps on the three ground types meant that the stored strain energy was not wasted by the somewhat useless extension arising from the slip or insertion of tibia. This finding coincides with the previous observation that the semi-lunar processes of locusts do not start to unfurl (i.e., release energy) until the tibia extends by 55 degrees [24]. In other words, the specific mechanism underlying the storage and release of strain energy helps locusts achieve high jumping performance on different ground types because the energy does not begin to release during the initial extension, regardless as to whether the tibia slips on or inserts into the substrate. This property might enhance the ability of locusts to evade predators and thus their survival in different environments.

By contrast, the postural features of locusts among the three types of ground indicated that significant differences were primarily observed in the elevation angle at take-off ($\beta_{t}$) and the tibial angle at take-off (γ) (Figure 4). Sutton and Burrows [6] found that the angle $\beta_{t}$ was almost the same as the tilt angle of the femoral proximal femur–distal tibia line before tibial extension ($\beta_{e}$) when locusts jumped on wood. Similar results were obtained by our measurements in that the difference between the two angles $\beta_{t}$ and $\beta_{e}$ was only 1.5 ± 0.83 degrees for jumps on wood. By contrast, the angle $\beta_{t}$ was significantly higher than the angle $\beta_{e}$ for jumps on the other two ground types (sand and soil). The differences between the three ground types stem from the specific conditions of the tibia on the ground (i.e., sliding or insertion). It was clearly illustrated in Figure 6 that the locust tibia firmly adhered to wood but often inserted into sand or slid on soil during the take-off process. The sliding or insertion altered the swinging angle of the hind leg, resulting in an altered take-off elevation angle. Although locusts can use their claws and tarsal adhesive pads to firmly stick to a few types of substrates [25,26], these strategies appear to be useless for successfully attaching to natural ground types, such as sand and soil. In addition, the significant difference in the angle γ resulted from the different $\beta_{t}$ among the three types of ground following the quantitative relationship between these two angles established by Sutton and Burrows [6].

Another interesting finding is that only the aft component of the mass-specific reaction force ($F_{a}$) was significantly different among the three ground types while its total magnitude and other components were not (Figure 5). This finding might stem from the differing effect of the tibial angle on the three physiological components. Given that the total reaction force ($F_{t}$) is along the tibial axis, its aft and normal components were calculated by the decomposition theory of force as $F_{t}\cos(\gamma )$ and $F_{t}\sin(\gamma )$, respectively. Thus, a higher tibial angle leads to a reduced aft component and an elevated normal component. Although the $F_{t}$ values did not significantly change among the three ground types, the aft component was increased by $0.35F_{t}$ (from $0.37F_{t}$ to $0.72F_{t}$) when the angle γ changed from 68 degrees (on wood) to 44 degrees (on sand) and by $0.26F_{t}$ (from $0.37F_{t}$ to $0.63F_{t}$) when the angle γ changed from 68 degrees (on wood) to 51 degrees (on soil). Similarly, the normal component decreased by $0.24F_{t}$ (from $0.93F_{t}$ to $0.69F_{t}$) when the angle γ changed from 68 degrees (on wood) to 44 degrees (on sand) and by $0.15F_{t}$ (from $0.93F_{t}$ to $0.78F_{t}$) when the angle γ changed from 68 degrees (on wood) to 51 degrees (on soil). Briefly, the influence of the angle γ on $F_{a}$ was 1.5–1.7 times that of $F_{n}$, which might explain why significant differences only existed in $F_{a}$ among the different ground types.

However, some limitations of this study require consideration. First, the sample size of jumps for each ground type was small, although paired statistical analyses were used in the comparisons. It should be emphasized that the results of this study are preliminary and should be used as a basis from which future studies examining the effect of ground type on locust jumps could be conducted. Second, the properties of the three ground types need to be quantitatively measured. Based on these measurements, the quantitative relationship between ground properties and locust jump behaviors could be characterized and provide some fundamental mechanical data for optimizing the jumping tactics of bioinspired robots on different ground types. 

## 5. Conclusions

In this study, jumps of nine adult *L. m. manilensis* locusts on three different types of ground (sand, soil, and wood) were measured using a custom-developed test system. Specifically, measurements were made of the postural features, kinematics, and reaction force. Both the elevation angle $\beta_{t}$ and tibial angle γ at take-off were significantly different among the three types of ground, which might have been caused by the hind legs slipping on or inserting into the ground. Nevertheless, the jumping kinematics (including the take-off speed and the mass-specific kinetic energy) were not significantly different among the different ground types, indicating that locusts were able to achieve robust jumping performance on the various substrates. This study provides preliminary data that contribute to enhancing our understanding of the jumping mechanisms in locusts, especially how jumping behaviors are adapted to different types of ground. 

## Figures and Tables

**Figure 1 insects-11-00259-f001:**
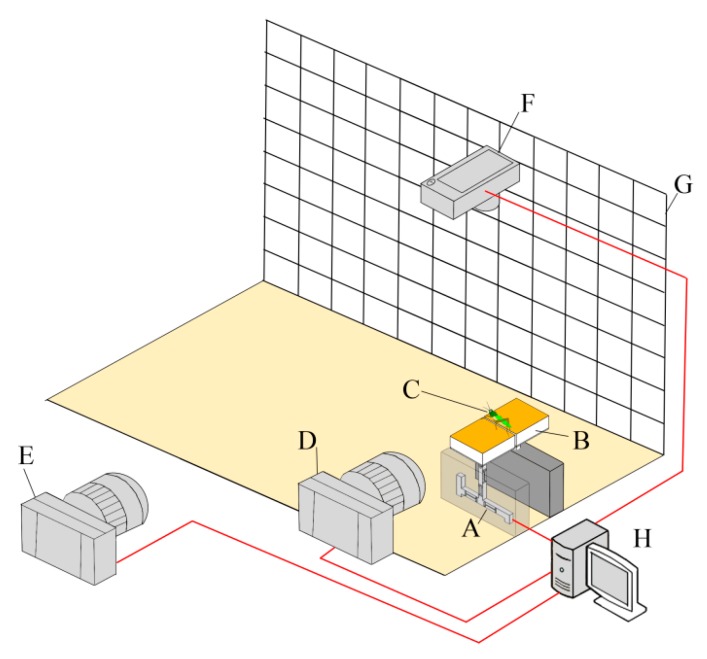
A custom-developed test system for measuring the kinematics and reaction force of adult locusts while jumping. (**A**) Three-dimensional high-precision force sensor; (**B**) two platforms, one of which is fixed to the force sensor and the other fixed to a stand column; (**C**) adult locust whose left and right hind legs each stood on one platform; (**D**) a high-speed camera for obtaining the rapid movement of the hind leg during the jump; (**E**) a high-speed camera for obtaining the trajectory of the locust jump; (**F**) a video camera for imaging the posture of the locust from above; (**G**) a coordinate background plate; and (**H**) desktop computer for data acquisition.

**Figure 2 insects-11-00259-f002:**
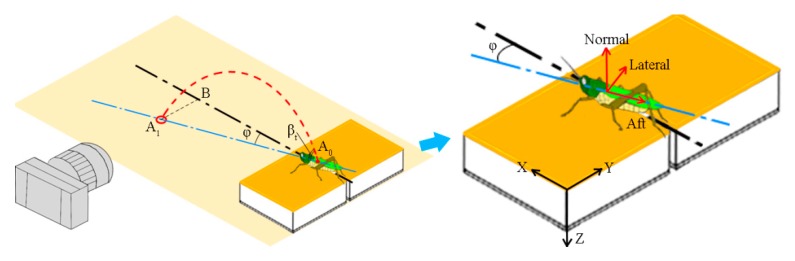
Schematic diagram illustrating how the kinematics and reaction force of the locust jumps were calculated. Because of the yaw angle (φ), the real trajectory of the jump was corrected from the image recordings of the sideway high-speed camera. The jump direction and real trajectory of the locust are indicated by the blue and red dashed lines, respectively. The black dashed line is parallel to both the lateral camera and the direction *X* of the force sensor. A_0_: the initial locust position; A_1_: the real locust position after jumping; B: the locust position in the camera images after jumping; and $\beta_{t}$: the elevation angle of the locust at take-off. Similarly, the physiological components of the reaction force were transformed from the sensor measurements using the yaw angle. The coordinate system of the force sensor is shown by black axes, and the physiological coordinate system of the locust is shown by red axes.

**Figure 3 insects-11-00259-f003:**
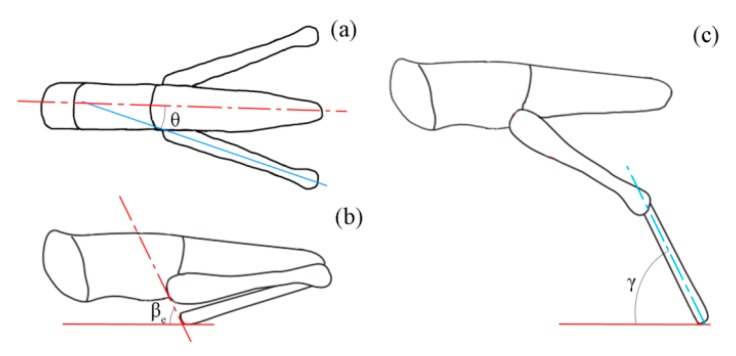
Schematic diagram for three postural features of adult locusts while jumping. (**a**) The opening angle between the hind femur and central line of the body just before tibial extension (θ); (**b**) the tilt angle of the line through the proximal femur and distal tibia just before tibial extension ($\beta_{e}$); and (**c**) the tibial angle relative to the ground at take-off (γ).

**Figure 4 insects-11-00259-f004:**
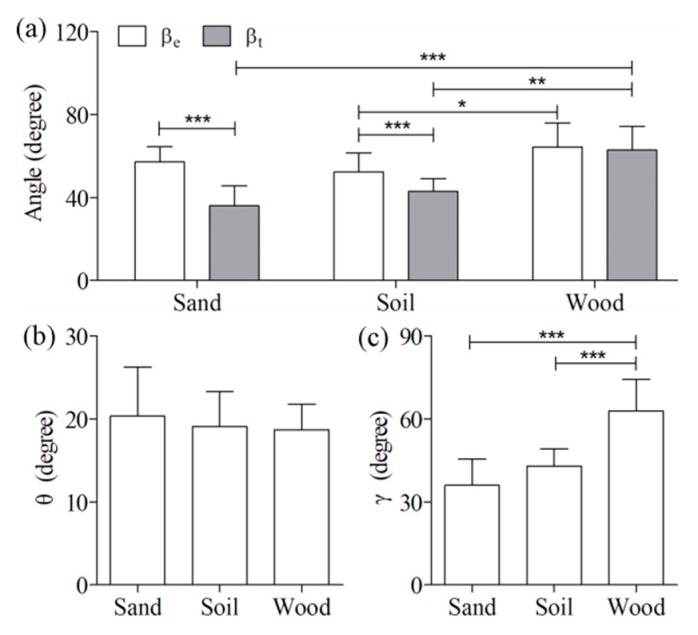
Comparisons of the postural features of adult locust jumps on the three ground types: (**a**) the elevation angle of locust jumps at take-off ($\beta_{t}$) and the tilt angle of the line through femoral proximal and tibial distal ends just before tibial extension ($\beta_{e}$); (**b**) the open angle of the femur to the central line of the body (θ); and (**c**) the angle between the tibial axis and the ground at take-off (γ). Results were reported as mean ± SD. * *p* < 0.05; ** *p* < 0.01; *** *p* < 0.001.

**Figure 5 insects-11-00259-f005:**
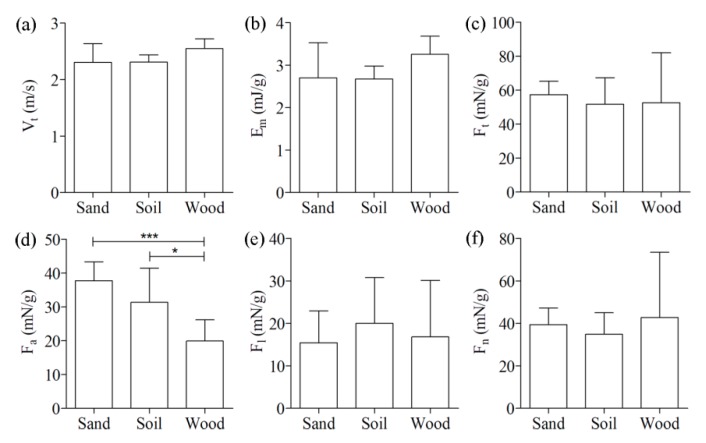
Comparisons of the kinematics and reaction force of locust jumps on three ground types: (**a**) the take-off speed ($V_{t}$); (**b**) the mass-specific kinetic energy ($E_{m}$); (**c**) the total magnitude of the mass-specific reaction force ($F_{t}$); (**d**) the aft component of the mass-specific reaction force ($F_{a}$); (**e**) the absolute lateral component of the mass-specific reaction force ($F_{l}$); and (**f**) the normal component of the mass-specific reaction force ($F_{n}$). Results were reported as mean ± SD. * *p* < 0.05; *** *p* < 0.001.

**Figure 6 insects-11-00259-f006:**
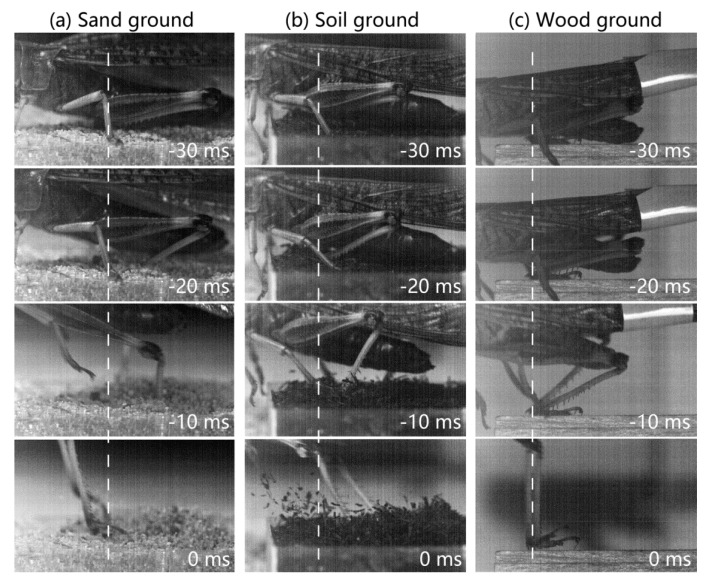
High-speed camera images for the rapid movement of the hind legs of locusts while jumping. Locusts jumped on sand (**a**), soil (**b**), and wood (**c**) ground types. The frame at 0 ms shows the configuration of the hind leg at take-off. To analyze the jump process, white dashed lines were marked for each panel to represent the location of the tibial distal end at 0 ms in the view of the camera.

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
