# Peer review of "The Effect of Ground Type on the Jump Performance of Adults of the Locust Locusta migratoria manilensis: A Preliminary Study"

_insects, 2020, doi:10.3390/insects11040259_

Round 1
Reviewer 1 Report
This paper reports on experiments to measure how the jumps of locusts are affected by the type of ground they are standing on. In particular, it was found that a softer type of ground (soil or sand) caused a reduction in the angle between the insect’s body and the ground at the time of jumping, leading to a higher force in the fore/aft direction. Other factors, such as jump velocity, were unchanged.
This is a useful contribution to our understanding of insect jumping behaviour. The experiments seem to have been done carefully and the results are clearly presented.
The English contains many small grammatical errors which will need to be corrected in the final version, but it was sufficiently good to convey the technical meaning of the work most of the time. There were however some cases where I was not sure what the authors meant, as a result of problems with the English: these cases are covered in the detailed comments below.
INTRODUCTION
1. Line 38 “scale-independent” : maybe you meant to say “scale dependent” since the following text says that there is a difference between juvenile and adult locusts.
- Line 48 “not rationale”: maybe you meant to say “not rational”, which has a different meaning. However I think on the contrary that it is rational to study the effect of roughness on solid ground, as the cited authors did. What you could say here is that there are other factors as well as roughness, which have not been studied.
MATERIALS AND METHODS
- It would be good to include more information about the three surfaces used. The test of a Methods section is whether the reader could exactly replicate the experiments as described. For this we would have to know some more about the various surfaces. For example, how was the surface of the balsa wood prepared: was it cut with a saw, sanded, etc? What was its roughness? Also more details about the sand and soil. There are many different types and grades of sand, with different grain sizes, etc. And how was the surface made flat in the case of the sand and soil surfaces – were they compressed for example? All these details could make a big difference to the nature of the surface in question.
- Line 101: Please say a little more about how the mass-specific kinetic energy was calculated. I suppose this is the mass/2 multiplied by the square of the velocity at the moment of take-off, divided by the mass. So therefore it’s simply half the velocity squared. How exactly is the velocity at that specific moment calculated? This is not a trivial thing to do accurately. It might be useful to show an example of the data coming from the equipment and how it is treated.
RESULTS
- Most journals now allow authors to include supplementary files such as videos, and this would be an excellent example of their value. The authors could upload typical videos showing take-off from each of the three surfaces, to illustrate their finding of how the end of the leg moves in the soft ground.
- Most journals now ask for all original experimental data to be made available to the readers. I would strongly encourage the authors to do this as their tests could be usefully reanalysed by other workers.
DISCUSSION
- The authors commented that though the angle (and therefore also the Fa force) changed with ground type, nevertheless this did not affect the jumping characteristics (velocity and energy). Presumably though it would have affected the jump distance. Probably the authors didn’t measure jump distance in this study but they might like to comment on how it would have been affected, which I think is simply a matter of doing a mechanics calculation.
CONCLUSIONS
- This section seems to just repeat information already given in the paper. It would be better to rewrite it, focussing on what exactly can be concluded as a result of this work.
Author Response
Dear Reviewer,
Thanks for your comments and suggestion. The point-by-point response has been listed in the attached file.
Best regards
Chao Wan

Reviewer 2 Report
“The effect of ground type on jump performance of adult locust (Locusta migratoria manilensis)”
The manuscript by Wan et al., submitted to “Insects”, studies the influence of different substrate types on locust jump performance. These jumps of locusts are important in the insect’s behaviour for movement and escape. Using a quantitative analysis, the authors find very similar jumping performance in adult locusts on three different substrates tested. The manuscripts’ main finding are differences in two parameters of leg movement and the reaction force for jumping movements on wood substrates.
The topic of the study is certainly of interest and worth investigating in detail, and the results as presented here point in an interesting direction, as the similarities found indicate great stability of the jumping movement sequence.
However, in the present form of the manuscript, the study appears rather preliminary in several aspects, outlined below. In sum, I recommend a rejection with option for resubmission after addressing the open issues:
The sample size is rather small with 9 animals. Although repeated measures are declared, the sample size for analysed jumps (N) on the different grounds and between sexes is nowhere given. This should be included for all the measurements which are analysed for mean values and statistics.
The study is very limited in its scope to test three different substrates for jump behaviours. Despite the intention to use such data for the design of bio-inspired insect-like robots, there is no biological concept that shapes a hypothesis as the basis for the animal choice, substrate choice, or experiments. Are locusts the best model organisms for insect-like robots? Do you assume those will be built with the intention for a capacity to jump, and why?
There are studies which demonstrate a significant influence of substrate type on behaviour for species of Orthoptera (e. g. Sarmiento-Ponce et al. 2018 Substrate texture affects female cricket walking response to male calling song, R Soc Open Sci 5:172334). Another aspect is the choice of tested parameters, which could also include e. g. tarsal angle (Witney and Hedwig 2011 Kinematics of phonotactic steering in the walking cricket Gryllus bimaculatus (de Geer), J Exp Biol 214: 69-79).
The introduction should develop the topic from a clear hypothesis, based on the functional requirements of jumping physiology.
The authors declare that certain roughness values of a substrate used in experiments cannot match the natural substrate, and list factors like stiffness, hardness, tangential friction and possible others (“etc.”) (l. 48 – 51). This is certainly correct, but then one would ask for more details on the natural substrates and those measured here, probably in form of a table which gives numerical values for the different materials to show the differences.
The statistical treatment with t-test and ANOVA are employed but no prior testing of data that the requirements for these tests are met were conducted.
The introduction opens with a focus on the different ground types, but the MS repeatedly stresses the possible differences between sexes (l. 15, 144, 193), including the opening of the results - the focus here should be consistent. Given the size and weight differences between sexes, did you compare the variations in the different data from females and males?
For the results, please present first the results from the experiments with reference to figures, and build the statistical analysis on top of that, not the other way around (results, l. 114 – 129). Figure 5 should be incorporated into results as it shows data, and not appear only in the discussion section.
Could you include the data on the slips and insertions, respectively, by giving percentages how often this occurred, depending on ground material? (compare l. 178) This information should also go into the results alongside Figure 5.
With respect to the difference in mass-specific kinetic energy, which is lower in wood – do you have also data on the distance of the jump? Does it differ between wood and the other ground types, or is it equal? Could this lower value be part of an adaptive jump initiation on a harder substrate?
At several points are the descriptions given unfortunately unclear. E. g. it is not explained what the “fore-aft component” of mass specific reaction force is and how it was measured; the tibial angle is not explained in the methods section, or what the authors mean by a neuron pattern. All the information given in the figures should be explained in the figure legend, like stating the meaning of symbols for angles in Fig. 2, or explaining what the bars in Fig. 3 mean (Standard error? Standard deviation?). One has to go back and forth between figures and text sections to look for explanations which distracts from the findings of the study.
Finally, the authors should seek a native speaker to check the MS for overall clarity. I am myself not a native speaker either, but below I give few suggestions on the abstract where the expressions are sometimes rather awkward:
Locusts are insects recognised for their outstanding jumping abilities. The jump performance depends on several factors, which include both the insect’s physiology and the natural environment. However, few studies have been carried out to investigate the effect of different ground types on the jump performance of locusts. Here, adult locusts (Locusta migratoria manilensis) were measured using a custom-developed test system to test their jump performance (including analysis of kinematics, body posture [specify further?], and reaction force) on three kinds of ground (sand, soil, and wood). The jump performance of adult female insects was not significantly different to adult males for all the three ground types tested. Significant differences only existed in single aspects of the legs elevation angle at take-off and a component of mass- specific reaction force [explain shortly] between wood and sand or soil. Slip on or insertion [of what?] into the ground was observed when the locusts jumped on sand and soil. Comparing different parameters of jump initiation did not reveal differences among the three kinds of ground, indicating a very robust locomotion sequence. This study could contribute to the bioinspired design of insect - like robots to achieve successful take-off on various substrates.
Author Response

(The authors gave the same response as above.)

Reviewer 3 Report
Dear Authors,
this study on the physical characteristics of locust jumping behavior is interesting, but needs considerable improvement in order to be suitable for a publication in “Insects”. I suggest a revision of the manuscript, based on the following points (see also comments in PDF):
- In general, please ensure correction of English by a native speaker or professional language editor (some minor corrections of grammar and spelling have been made in the PDF document).
- “Insects” has readers mainly from entomology, so it would be beneficial to adapt the text so as to shift the focus from physics and robotics to more biological topics, e.g. fitness, predation avoidance, optimal foraging... and include relevant references in the Introduction and Discussion.
- L 16-18: Is this your key finding? Please highlight throughout the manuscript and especially in the abstract, what might explain this significant difference and why this result is interesting / new / useful.
- L 21-22 (Abstract), L54-55 (Introduction), L201-202 (Conclusion): How is this repeated mention of robots relevant to the study? What is their use? Please also elaborate some other potential uses or fields for further study based on your results.
- L61-72: Refer to the letters A-H from Fig. 1 within this text section, to improve understanding.
- L83: Can you rule out that taping the wings may change overall movement pattern or introduce unnatural stress to the animals, thereby influencing your results?
- L84: Why did you use two different triggers? Were reactions comparable?
- L 90: “repeatedly” = how often? This is important here regarding the very small sample size. Is there a way to include data from more specimens or justify why such a small sample was used? With such a small sample, I would rather call it a "preliminary study" as a basis for further analyses.
- L92-93: As this is not a physics journal, please explain this section in more simple terms suitable for biologists.
- L99: Define “triggering” in this context.
- L105 (Fig. 2): These schematic images are nice and improve understanding - if possible recreate them with a professional drawing program.
- L157 (Fig. 5): Is it possible to get higher resolution and/or color images? This is interesting but hard to see.
- L164-173: These calculations represent further results and should be moved to the Results section, including an easily understandable explanation for entomologist readers. In the Discussion, please explain what this means in biological terms, regarding fitness, energy expenditure, predation avoidance etc. for the locust?
- L178: “unexpected” - was it really unexpected? Soil and sand is less stable than wood.
- L183-184: This is very interesting - please elaborate more on the energy conservation aspect, also in Abstract and Introduction.

Author Response

(The authors gave the same response as above.)

Round 2
Reviewer 2 Report
This revised version of the manuscript is now thoroughly corrected and improved in the overall presentation. It presents the data obtained from jumping experiments in a much more coherent way and flows better in this version. Additional quantitative data (on tarsal slipping and tibial insertion) give a more thorough picture of the jump behaviour, the statistics description is more complete, and the organisation of the material (results - discussion) are now improved. The language has been carefully edited, and the manuscript gains considerably to communicate its findings.
Regarding the comments from the previous reviewing, I would like to make few more points clearer, which could be addressed for further correction before acceptance for publication:
l. 32 do you mean: “patterns of muscle and motoneuron activity”?
l. 63 I did not mean to criticise your work here as preliminary, but maybe there are some values for material properties of the substrates from the literature that could be cited to give the reader a quick overview of the differences?
l. 112 the number of jumps evaluated overall is still not spelled out in the main manuscript (n = 9; N = ?). This remains problematic, as you include in the results a number of nine jumps on sand and soil (l. 203-204). But you declare that each locust individual was analysed for repeated jumps? These analysis should basically state three additional numbers, one for the overall number of jumps analysed on each ground type from the 9 insects, and these could go here into the method section, or later in the results section.
l. 68/ 221 This may be a matter of perspective, but as the introduction assumes to find major differences in jumps depending on the ground type, one could argue that most components of the jumping are actually highly similar, indicating a very stable locomotor sequence, which is not strongly different between substrates. This robustness only comes up at line 242 in the discussion, and might serve as a good opening of the discussion, as it is a central take home-message from the jump analysis.
Author Response
Dear reviewer,
The point-to-point response to the comments has been listed in the attached file. Please see the attachment.
We gratefully thank for your review on our manuscript.
Bests regards
Chao Wan

Reviewer 3 Report
Dear Authors,
Based on the changes made to the manuscript and the replies to reviewer comments, the paper is now suitable for publication in my opinion. Especially the professional language editing and the newly designed figures improve the quality greatly.
While still heavy on physics and mechanics content, the results are now presented in a way that is also understandable and of interest to the entomological community. Calling it a preliminary study in the title puts the results into adequate perspective and justifies further interesting research.
Please just make the following small corrections to the supplementary tables: “jumped” -> “jumping”; “Bodymass” -> “Body mass”

Author Response
Dear reviewer,
We gratefully thank for your review on our manuscript. Our response to the comment has been listed at the end.
Bests regards
Chao Wan
Q1: Please just make the following small corrections to the supplementary tables: “jumped” -> “jumping”; “Bodymass” -> “Body mass”
A: We have modified the supplementary table legends accordingly.